# Unveiling biogeographical patterns of the ichthyofauna in the Tuichi basin, a biodiversity hotspot in the Bolivian Amazon, using environmental DNA

Cédric Mariac[1,2]*, Fabrice Duponchelle[2,3]ᵒ, Guido Miranda[2,4,5]ᵒ, Camila Ramallo[4], Robert Wallace[4], Gabriel Tarifa[4], Carmen Garcia-Davila[2,6], Hernán Ortega[7], Julio Pinto[5], Jean-François Renno[1,2]

1 DIADE, Univ Montpellier, CIRAD, IRD, Montpellier, France, 2 Laboratoire Mixte International—Evolution et Domestication de l'Ichtyofaune Amazonienne (LMI—EDIA), IIAP, UAGRM, IRD, Paris, France, 3 MARBEC, Univ Montpellier, CNRS, Ifremer, IRD, Montpellier, France, 4 Wildlife Conservation Society, Bolivia Program, La Paz, Bolivia, 5 Unidad de Limnología, Instituto de Ecología, Universidad Mayor de San Andrés, La Paz, Bolivia, 6 Instituto de Investigaciones de la Amazonía Peruana (IIAP), Laboratorio de Biología y Genética Molecular (LBGM), Iquitos, Perú, 7 Departamento de Ictiología, Museo de Historia Natural, Universidad Nacional Mayor San Marcos, Lima, Peru

ᵒ These authors contributed equally to this work.
* cedric.mariac@ird.fr

**Data Availability Statement:** Sequencing reads were deposited in the National Center for Biotechnology Information Sequence Read Archive

## Abstract

To date, more than 2400 valid fish species have been recorded in the Amazon basin. However, some regions remain poorly documented. This is the case in the Beni basin and in particular in one of its main sub-basins, the Tuichi, an Andean foothills rivers flowing through the Madidi National Park in the Bolivian Amazonia. The knowledge of its ichthyological diversity is, however, essential for the management and protection of aquatic ecosystems, which are threatened by the development of infrastructures (dams, factories and cities), mining and deforestation. Environmental DNA (eDNA) has been relatively little used so far in the Amazon basin. We sampled eDNA from water in 34 sites in lakes and rivers in the Beni basin including 22 sites in the Tuichi sub-basin, during the dry season. To assess the biogeographical patterns of the amazonian ichthyofauna, we implemented a metabarcoding approach using two pairs of specific primers designed and developed in our laboratory to amplify two partially overlapping CO1 fragments, one of 185bp and another of 285bp. We detected 252 fish taxa (207 at species level) among which 57 are newly identified for the Beni watershed. Species compositions are significantly different between lakes and rivers but also between rivers according to their hydrographic rank and altitude. Furthermore, the diversity patterns are related to the different hydro-ecoregions through which the Tuichi flows. The eDNA approach makes it possible to identify and complete the inventory of the ichthyofauna in this still poorly documented Amazon basin. However, taxonomic identification remains constrained by the lack of reference barcodes in public databases and does not allow the assignment of all OTUs. Our results can be taken into account in conservation and management strategies and could serve as a baseline for future studies, including on other Andean tributaries.

(BioProject ID: PRJNA661268, BioSample accessions: SAMN16049263- SAMN16049300).

**Funding:** CGD has been supported by Programa Nacional de Investigación Científica y Estudios Avanzados (PROCIENCA) [grant number 017-2018-FONDECYT]. GM, RW, GT, CR has been supported by Gordon and Betty Moore Foundation [grant number GBMF331.07]. CM, FD, JFR, CDG has been supported by Laboratoire Mixte International - Evolution et Domestication de l'Ichtyofaune Amazonienne (No grant number). The funders had no role in study design, data collection and analysis, decision to publish, or preparation of the manuscript.

**Competing interests:** The authors have declared that no competing interests exist.

## Introduction

The Amazon basin hosts the largest freshwater fish diversity on earth with 2,406 native species [1, 2] as well as a high genetic diversity [3], but is facing increasing threats from anthropogenic activities such as deforestation, habitat degradation, overexploitation, hydropower dam building, invasive species and pollution, and from climate change [4–6]. Yet, the high number of new fish species described every year suggest that, as impressive as it already is, the species list of the Amazon basin is far from complete [7]. The Madeira is one of the most species-rich basins of the Amazon [2]. It includes the Beni watershed, already considered vulnerable to anthropic pressure and where knowledge gaps remain, such as in the Tuichi River [1]. The Tuichi basin is part of the Madidi National Park and Natural Area of Integrated Management (hereafter abbreviated Madidi NP), which is included in the Tropical Andes hotspot of biodiversity [8–10] and also hosts one of the few identified spawning area for many fish species [11]. It therefore appears urgent to implement conservation and management measures to protect this biodiversity and to fill known gaps in terms of collection and knowledge of fish species [1].

Recently, environmental DNA (eDNA) sampling has been developed as an efficient and reliable method for understanding the biodiversity of all taxonomic groups in an ecosystem [12–15]. Environmental DNA sampling is a rapid and non-destructive method. It can target small taxonomic groups and even detect rare or cryptic species found at low abundances. For example, in the Brazilian Atlantic forest, a single eDNA sampling detected the nine known species of Anuran, from the most abundant to the rarest, demonstrating the sensitivity of the approach [16]. In herpetofauna, eDNA sampling detected different species of turtles and amphibians in lakes and rivers in North America including the rarest species [17]. In the streams of central Idaho rivers, the giant salamander (*Dicamptodon aterrimus*) and the Rocky Mountain tailed frog (*Ascaphus montanus*), two rare species of amphibian, were detected thanks to eDNA sampling [18]. The eradication of an invasive species often requires early detection, and eDNA has proven more effective than traditional methods for detecting introduced buffalo toads in France [19], and the python, *Python bivittatus*, in Florida [20]. Similarly, species that were believed to be extinct or residual in a marine ecosystem, including large shark species, were detected using the eDNA approach, thereby facilitating the development of conservation strategies [21]. Aquatic mammals can also be detected: three endangered species of manatees (*Trichechus ssp.*) have been successfully detected in African, Amazonian and Indian rivers, helping to refine known distributions and establish conservation management strategies [22]. eDNA can also complement acoustic conservation monitoring for marine cetaceans in vast oceans [23]. So far, however, the eDNA method has been more rarely used to assess general biodiversity, especially in extremely biodiverse areas [24]. The Neotropics, which harbour about 30% of all described freshwater ichthyofauna, is one such extremely biodiverse area for fish species [7, 25]. Very few eDNA studies were carried out so far on the ichthyofauna of the Neotropics (e.g. [25–28]), and even fewer in the most species-rich basin of the Neotropics: the Amazon basin [27, 28]. All these studies were limited by the size of the reference database. Nevertheless, with suitable database development, eDNA will revolutionize research on rare or very discrete species, but also our understanding of biodiversity, both crucial aspects for the sustainable management of aquatic ecosystems.

Here, we tested the effectiveness of eDNA assessment (1) to fill the knowledge gap regarding the ichthyological diversity in the hydrological basin of the Tuichi River [1], hypothesizing that eDNA would be more efficient than traditional assessment methods for a quick inventory of species diversity and to unravel rare or cryptic species, and (2) to investigate the biogeographical structure of the Tuichi's ichthyofauna. The information provided will be used for

the management of the Madidi NP and serve as a reference for the management of other hydrographic basins in the tropical Andean foothills.

## Methods

### Sampling

**Study area and collecting sites.** The sampling area is located mainly in the Tuichi sub-basin in the Beni watershed, sub-basin of the Madeira River, the largest tributary of the Amazon (Fig 1). It is part of the Madidi NP, one of the largest protected areas in Bolivia covering 1,895,750 ha [29]. The Tuichi sub-basin constitutes a hydrological network of more than 7,100 km (including all tributaries) that extends into the Andes at an altitude of more than 6,000 m a.s.l. and flows 270 km to its confluence with the Beni River, 25 km upstream of the town of Rurrenabaque. Environmental DNA samples were collected from lakes and rivers between September and October in 2017 (2 sites), 2018 (10 sites) and 2019 (22 sites); each site being sampled only once (see S1 Table for details and geographical coordinates). This period of the dry season with low water levels allows for easier navigation. But more important, during this period the connectivity between lakes and rivers is interrupted, which ensures that eDNA collected in the lakes provide information on their own ichthyofauna without eDNA input from the main stream. In the Tuichi basin 22 sites were collected: 12 on the Tuichi River, eight on its main tributaries and two on tectonic lakes (Chalalán and Santa Rosa). Additionally, in order to assess a possible peculiarity of fish compositions in the Tuichi basin, we also sampled two sites on the Beni River, one upstream and one downstream of the confluence with the Tuichi, three tributaries of the Beni (Hondo, Quendéqué and Quiquibé rivers) upstream of the confluence with the Tuichi, one oxbow lake (Ruta) on the Hondo River, as well as 6 other lakes located downstream in the Beni basin, outside of the Madidi NP. A total of 34 sites were collected, 25 in rivers and 9 in oxbow lakes (Fig 1A and 1B) covering an altitudinal range from 200 to 4,505 m a.s.l. The average river network distance between sites collected in the Madidi NP is 14.4 km (SD = 21.8). Overall, the time required to complete eDNA collections was approximately four weeks. Following an ordering system of drainage patterns [30], the Beni River, the main river in the watershed, receives the rank 1; the Tuichi, Quiquibé, Quendéqué and Hondo rivers were put in rank 2 and all the Tuichi's tributaries in rank 3 (Fig 1).

According to Wasson et al. (2002) [31] geomorphological and climatic structures delineate nine geoclimatic domains sub-structured into 40 hydro-ecoregions (HER2) in the Bolivian Amazonian basin. HER are congruent with geology, relief, climate (temperature and precipitation), vegetation, soils and hydrography. Our sampling sites are distributed across 7 HERs (Fig 1 and S1 Table for details) corresponding to Glacial mountain (Cg, 1 site), High peri-Amazonian Yungas (Ya-a, 1 site), Amazonian gallery forest (Ab-a, 2 sites), Beni Gallery Forest (Ab-b, 4 sites), Dry valleys of the Yungas (Ay, 2 sites), High humid sub-Andean (Sa-a, 1 site) and Low humid sub-Andean (Sa-b, 23 sites).

*Ethic statement*. The Wildlife Conservation Society (WCS) is authorized by the Bolivian government to study and sample the Bolivian Amazonian biodiversity. The Ministerio de Medio Ambiente y Agua (MMAyA) of Bolivia approved the study and provided permits (autorization numbers: FCPV-IE-0196/2017; FCPV-IE-0030/2018; FCPV-IE-0064/2019) for collection, export and analysis of the filter cartridges to the French National Institute of Research for Sustainable Development (IRD) laboratories in France.

*eDNA field sampling*. Due to variations in water turbidity in these areas, particularly in streams, and in order to ensure sufficient filtered volume before clogging, we used sterile Envirocheck HV filter cartridge with a porosity of 1 µm (Pall Corporation, Ann Arbor, MI, USA). One cartridge per river sampling site was used, and two to five cartridges for lakes. For

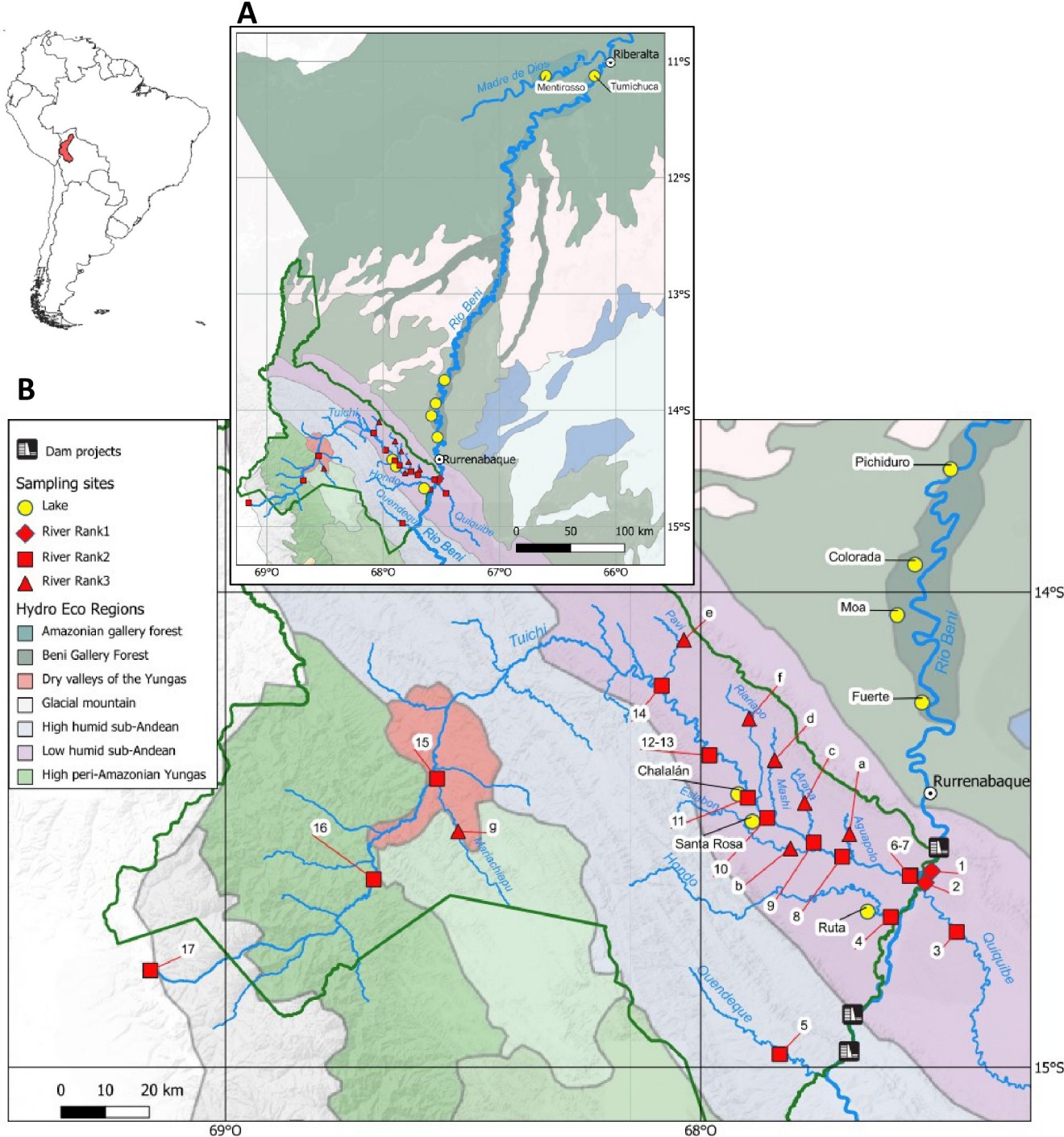

**Fig 1. Map of the sampling locations.** Map A shows the location of the 34 sites sampled and Map B focuses on sites south of the 13th parallel. The lake sites are represented by yellow disks (labelled with their full name), river sites are represented according to their hydrographic rank 1, 2 and 3 by red diamonds (samples 1–2), red squares (samples 3–17) and red triangles (samples a-g), respectively. Samples 6 and 7, as well as 12 and 13, were carried out at two sites on the Tuichi river in 2018 and 2019, respectively. Maps were created with QGIS v3.4 using the base map Stamen™ Terrain Background. The points were shifted for better viewing, but the actual positions are reported in the S1 Table. The hydro-ecoregions (reprinted from Wasson et al. 2002 under a CC BY license, with permission from International Association of Hydrological Sciences, original copyright 2002) were also indicated (see legend for details).

river sites where the width of the river was greater than 20 m, sampling was carried out by boat at three points, one in the middle and the other two a few metres from each bank. Otherwise a single point was sampled in the high-flow part of the river. The average sampling depth is 1.15 m SD = 0.6 (see S1 Table for details). The sampling points on the tributaries of the Beni and Tuichi rivers were carried out at least 200 m from their confluence. In lakes, sampling was carried out by boat and designed to integrate as much as possible spatial heterogeneity of eDNA distribution, as previously suggested in lentic ecosystems [32, 33]: collecting points were distributed over the entire area of the lake (see S1 Table). The total number of sampling points per lake ranged from seven for the smallest lake, Ruta, to 90 for lake Tumichuca (S1 Table).

The sampling kit is composed of a drill driver (AEG GmbH -BSB12cli202c), a peristaltic pump (Watson-Marlow 313-DW) in which the marpene suction hose was connected upstream to 5–6 m of a polyethylene suction hose and downstream to a polypropylene hose connection ensuring connection to the filter cartridge. Lead sinkers and a suction strainer (2 mm mesh) were added to the end of the suction pipe to ensure ballasting and prevent the entry of large particles into the cartridge.

All materials were handled with sterile gloves and between each site, tubings, hose fittings and suction strainer were decontaminated in a 9% sodium hypochlorite solution for 10 min [34, 35], then rinsed three times with commercial water. The driver unit was decontaminated using DNA away solution (Thermo Scientific™). In order to control possible contamination of equipment and cross contamination between sites, 7 negative control cartridges interspersed throughout the sampling campaign were made by filtering, for each cartridge, 20 l of commercial water in the field (see details in S1 Table).

Each eDNA sample was taken by filtration of water pumped directly into the medium, applying an initial flow of approximately 1 l/min. Filtration was carried out until the filter was clogged and during the whole sampling phase the cartridges were protected from light by aluminium foil. The final volume filtered for each cartridge varied between 6.5 and 75 l, depending on the clogging speed as a function of the turbidity level at each site. Total volume per site filtered ranged from 11 to 180 l.

At sampling completion, cartridges were filled with 150 ml of conservation buffer (Tris-HCl 0.1 M, EDTA 0.1 M, NaCl 0.01 M and N-lauroyl sarcosine 1%, pH 7.5–8) and stored at 4˚C until DNA extraction between 1–2 months after sampling. A total of 58 cartridges were processed, including seven negative field controls performed on average every 7 cartridges (SD = 2.8; commercial water filtered in the field) and one alien control consisting of four species (*Alburnus alburnus*, *Carassius gibelio*, *Cyprinus carpio*, *Gambusia holbrooki*) never recorded in our study area. This control was collected in an ornemental concrete pond at IRD Montpellier (France), which contains only species not recorded in the Beni basin (Cypriniformes and Cyprinodontiformes), then none should be detected in our samples if cross-contamination and hopping index are controlled.

**Laboratory analysis.** *DNA extraction*. The DNA extractions and the PCR amplification steps were carried out in dedicated and physically separated rooms. To minimize the risk of cross contamination, all PCR steps were prepared under a workstation (GuardOne® StarLab) allowing UV decontamination.

Three independent extractions were performed for each cartridge as follows: 5.4 ml of a 2.5 M NaCl solution were added to each cartridge to obtain a concentration of 0.1 M and 3 g of SDS to a final concentration of 2%. After a 2 h 30 lysis at 56˚C with stirring, the cartridge was emptied into 10 tubes of 50 ml. An additional volume of 7.5 ml of 3M K acetate is added to precipitate the proteins. After centrifugation for 30 min at 4˚C at 6,200 g the supernatant was collected by pipetting to which 10 ml of isoamyl chloroform (24:1) was added, mixed by reverse (50 times), then centrifuged for 15 min at 15˚C and 6,000 g. The aqueous phase is

recovered by pipetting and then 1.7 ml of Na acetate 3M pH 5.5 and 11 ml of cold isopropanol were added and mixed by inversion to precipitate DNA, then stored at -20°C for 48h, centrifuged 30 min at 4°C and 6,200 g. After elimination of the supernatant, the DNA pellets are washed with 1.5 ml of 70% ethanol, centrifuged for 15 mins and allowed to dry after elimination of the supernatant. The pellets are finally resuspended in 200 μl of TE. A 25 μl fraction is purified with XP ampure beads (1.8X) before amplification by PCR.

*Primers design, PCR amplification of barcodes and library preparation*. We designed two COI metabarcoding primers targeting Amazonian fishes: one of 185 bp (MK1) and another of 285 bp (MK2) which share the same forward primer and are positioned 5' from the standard COI-5P barcode. The R Primerminer package [36] was used to extract the COI sequences available of the 10 main freshwater fish Orders present in the Amazon basin. A manual design of the primers was carried out on the 10 consensus alignments carried out for each Order. These two barcodes were amplified separately with the following primers: F1-TCHACHAAYC AYAAAGAYATYGGYACYCT with R1-ACYATRAARAARATYATYACRAADGC and F1 with R2-CARAARCTYATRTTRTTYATTCGNGG. Given the fact that eDNA samples are known to be generally quite fragmented (Harrison et al., 2019 [37] for a review and Deiner et al., 2017b [38] for contrasting results), we chose to use these two barcodes and compare their performance. The second barcode which completely overlaps the first is supposed to be more resolutive because it is longer but less sensitive given its length. The taxonomic resolution, at the species level, of MK1 and MK2 barcodes for Amazonian fish species was 85 and 89%, respectively (see details in S2 Text). The libraries were prepared according to a 2-step PCR [39]. Each library was indexed using unique combinations of double indexing to more effectively exclude chimeric amplicons and index hopping during downstream analysis (unexpected combinations were assigned as indeterminate). For each filter cartridge three libraries from three independent DNA extractions were amplified for both barcodes.

Two mock samples made up of 40 taxa (DNA extracted from tissues) for which we know the exact taxonomic composition were included to assess the sensitivity and specificity of the barcodes used in this study. As controls, they also make it possible to validate the bioinformatic treatments implemented, because we expected that the estimation of their specific composition would be close to the known composition.

An additional control called "alien" was included to verify cross-contamination during the preparation and sequencing of libraries. In addition to the 22 negative field controls (3 DNA per negative cartridge +1), 14 extraction negative controls and 6 PCR negative controls (ultrapure water instead of DNA) were also amplified and sequenced in the same run with the samples. Overall including negatives controls, positives controls and samples, 200 libraries for each barcode were pooled, then purified and sequenced (paired end, 150 bp) using Novaseq 6000 Illumina platform at the Novogene Co., Ltd. facilities.

**Data analysis.** *Data processing and taxonomic assignment*. Amplicon sequences were analysed using the FROGS pipeline [40] with standard operating procedures. Amplicons (MK1 and MK2) were split according to their primer sequences and size (120–280 nucleotides and 200–300) and clustered into operational taxonomic units (OTUs) using Swarm (aggregation distance: d = 1). After chimera removal, OTUs were kept if present in at least one library and with a minimum abundance of three sequences.

OTUs alignment against a custom COI fish database was performed by blastn (version 2.8.1). This local database was built with fish COI sequences mined from GenBank and Bold and also implemented with published sequences [41], but not yet available under GenBank (S2 Table). This final database has 160,944 entries representing 3,480 genera and 15,429 fish species distributed worldwide. For Peru and Bolivia among the 1,605 species currently known [42, 43] only 684 species are represented in the COI database. Although this demonstrates the

incompleteness of the COI database for Amazonian fishes, this barcode is currently the one with the best coverage in the number of species.

Blastn results were parsed using MEGAN software [44] in order to perform affiliation with a minimum score value of 150, a minimum percent identity of 97% and the naive LCA algorithm at minimum percent coverage of 50. The three types of negative controls (field, extraction and PCR) were used to identify and remove contaminating reads in sample libraries with the R package Microdecon [45], which uses the data in the 42 negative controls performed throughout the entire project to identify and exclude the reads in each library that correspond to contamination. The Maximum Sensitivity plus Specificity (MaxSSS) threshold value [46] was computed using the two positive controls (species composition known) and the R package ROCR 1.0–7 [47]. This minimum frequency at which a taxon is retained was applied in order to maximize the sum of True Positive Rate and True Negative Rate. To assess whether the sequencing effort was sufficient, the rarefaction curves for each library were performed with the R Vegan 2.5.6 package [48].

*Taxonomic diversity and geographic distribution.* Taxa richness (alpha diversity) and community composition (beta diversity) were computed with R packages phyloseq 1.28 [49] and heatmap3 1.1.6 [50].

To detect possible biogeographic entities, a hierarchical cluster analysis (HCA) was performed using the hclust function of the vegan 2.5.6 R package [48] to build a dendrogram clustering the sites according to their level of faunistic identity using Jaccard's binary distance and a Ward2 aggregation method. A bootstrap test with a number of 1000 replications was performed to evaluate the robustness of each node of the dendrogram using the package pvclust 2.2.0 [51]. A precise identification of a biogeographic structure with eDNA samples collected in the aquatic environment is only possible if its diffusion is spatially limited. We have therefore estimated, among our samples, the minimum distance at which species present at one site are not detected at the downstream site.

In order to assess the relevance of our sampling effort we computed the second order jackknife (Jack2) estimator [52] using the package Vegan with the functions specnumber and specpool and calculated the percentage of coverage as: observed number of species / Jack2 estimator) calculated for the entire study area and for each biogeographic entities defined by hierarchical cluster analysis. It should be noted that the Jack2 estimator (derived from the number of singletons and doubletons observed) only partially represents the actual species richness of the study area due to the incompleteness of the current COI database which contains only 42.6% of the known species in this Amazon watershed. Moreover, the species richness revealed in this study is also partial because our sampling was only carried out during the flooding period and therefore lacks species whose presence would be exclusively seasonal and outside this period.

*Environmental factors correlated to the biogeographic structure.* The link between environmental factors and the biogeographic structure of fish species was measured for four categories of variables noted at each site: 1) the geomorphological environment of the habitat, including the lake or river ecotype, the size of the lakes and their age, as well as the area of their hydrographic basin and the length of the hydrographic network that drains into them, 2) the geographical location, including latitude (LAT), longitude (LONG) and altitude (ALT), 3) the substrate, corresponding to the nature of the sub-soil and soil at the sampling point, according to five geological types (GEOL) and six pedological types (PEDO) and, 4) hydro-ecoregions (HER), which integrate some of the above variables and define 7 different categories for the sites sampled in this study (Fig 1). Lake sizes, hydrological networks length and basin area values were extracted for each site using Hydro Atlas [53] under QGIS version 3.4.12 [54] from GIS layers obtained at the World Data Bank (https://datacatalog.worldbank.org/dataset/

bolivia-soil-classification; https://catalog.data.gov/dataset/south-america-geologic-map-geo6ag) and from Schneider et al., 2017 [55] (S3 Table).

Non-metric multidimensional scaling (NMDS) using Jaccard's binary distance was performed and correlations between environmental variables and NMDS ordination were measured.

Considering that some variables are not independent, such as longitude and altitude, the objective of this analysis is therefore to identify environmental variables which are not necessarily explanatory but that can be correlated with the structuring of the ichthyofauna. Correlations between environmental variables and NMDS ordination were computed with the function "envit" in the R package Vegan. A similarity analysis (ANOSIM, package Vegan 2.5.6) was performed to test whether the biogeographic entities identified by the HAC were significantly different in the NMDS projection. Data analyses were completed in R version 3.6.2 [56].

All command lines and scripts used for bioinformatic treatment, statistical analyses and carrying out graphs are available in S1 Text.

## Results

### Sequencing, assignation and controls results

**NGS sequencing and taxonomic assignment results.**   Overall for the 200 libraries, which included 154 samples, two controls, one alien sample, 42 negatives (6 PCR-blank, 22 negatives field controls, 14 negatives DNA extraction), sequencing produced 551.8 million raw reads (82 Gb). The mean number of reads per site was 14.7 million (SE = 12.6), see S4 Table for details.

Five libraries with abundance below 50 sequences assigned to fish have been excluded from the analyses: they included all libraries (3) from one site on the Tuichi (not reported), and one library each for the Chalalán lake and the Tuichi site 14. Rarefaction curves done at the library and site level (S1 Fig) show that sequencing effort was enough for the remaining libraries.

After cleaning reads, 157.6 million could be joined and were clustered in 210,035 OTUs. Of these, 5,246 OTUs made up from 12.59 million sequences were assigned to a fish COI reference (median value per site = 170,742), 93.6% at the species level. OTUs (n = 8,502) with a match with a COI reference in the database, but with scores or percentages of identity values below the thresholds were discarded. Finally, 196,287 OTUs had no hit.

After taxonomic assignment, 120599, 1418, 54 and 7 reads were aligned to *Marcusenius monteiri* (Bold: AMNHI782-12), *Epinephelus marginatus* (GenBank: JX124778.1), *Zoarces gillii* (GenBank: HM180941) and *Sebastiscus marmoratus* (GenBank: HM180872), respectively. As they correspond to marine or non-Amazonian taxa, we verified these 4 sequences by blasting them under BOLD / NCBI. The results clearly show that the names of these four accession numbers are erroneous as their sequences show high blast score and % identity with bacterial or human origin. They were, therefore, discarded.

One hundred-two taxa were shared by the two primer sets (MK1 and MK2) and 90 taxa were unique to a single primer set (73 for MK1 and 17 for MK2). Although barcode MK1 is smaller in size than MK2 (185 and 285 bp, respectively) their taxonomic resolutions were similar, with more than 90.6% and 92.7% of the OTUs assigned at the species level. The percentage of identity between the OTU and the reference sequences of the database to which they are assigned was on average 98.71% (SE = 1.12). Taxa abundance per library are reported in S5 Table.

*Controls by negative, positive and alien samples.* Of the 42 negative control libraries only 213 reads were assigned to fish taxa indicating a low level of contamination. Of 40 species in the mock control samples, 39 have been recovered by our primers, which validates their

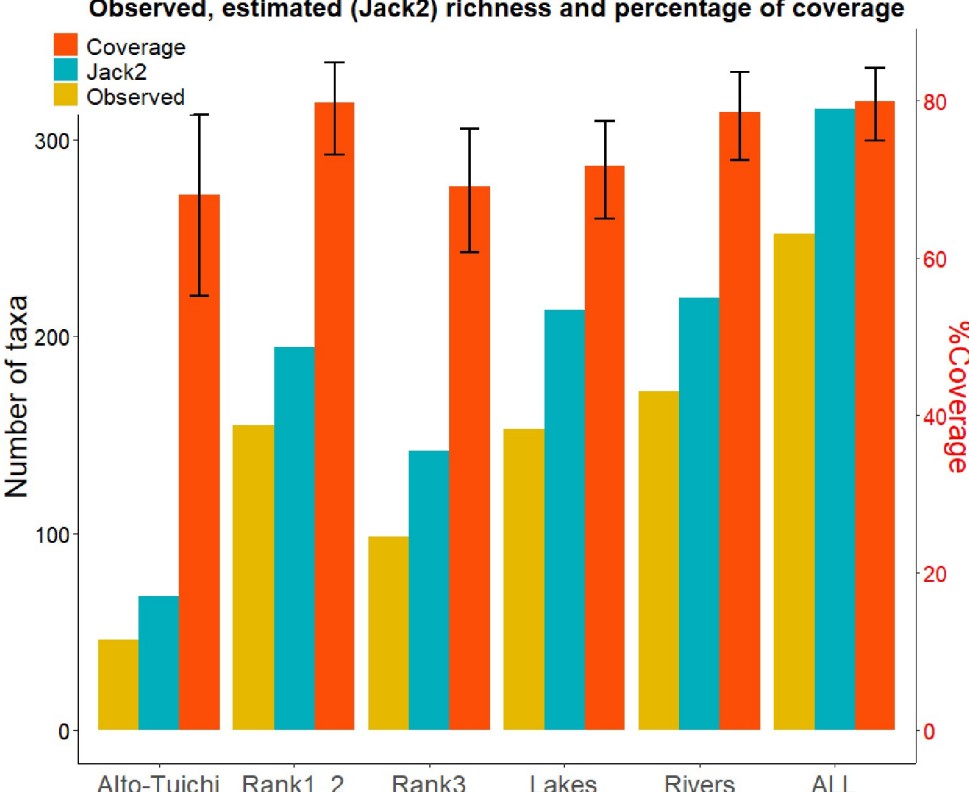

**Fig 2. Number of taxa observed (green bars) and second order Jackknife estimate of taxa richness (blue bars) computed per biogeographical entities, lakes, rivers and overall sites (given the COI database).** The orange bars represent the percentage coverage (ratio between the number of taxa observed and the Jack2 estimator) with black vertical error bars indicating the 95% confidence interval.

effectivenesswere identified. These mock samples were used to set a minimum frequency at which a present taxon was declared. This threshold was calculated to maximize the rate of true positive and to minimize the rate of false positive and set a minimum value of 0.01% (MaxSSS) from which a taxon identified in a library is kept. In the "alien" sample, (included to control index breaks), the four expected taxa (which are absent in the Amazon) were identified. A maximum of three reads belonging to a species from this alien control were found in an Amazonian sample, which shows a low level of leakage or contamination occurred during library preparation or the sequencing process. On the basis of this estimated leak rate, an additional filter was applied and only the taxa represented by at least four reads per site were kept.

**Biodiversity.** *Taxonomic diversity in the study area.* On all the sites, 252 taxa were identified covering 79.8% of the estimated number of identifiable taxa (Jack2 = 315) given the limitations of our database (Fig 2). Among these taxa, 22 are identified to the genus level, 207 at the species level and 23 at the level of undescribed species (noted sp.) and all are described in South America (FishBase, [57]. Among the 207 species identified 150 have already been identified in Bolivia [2, 43]. The 57 species never identified to date in Bolivia were verified on the basis of systematics and biogeography. Of these, 16 were named *affinis* (although they had a hit of more than 97% identity) because they were never identified in the Amazon basin. Among these 16 species, 11 have an abundance of more than 193 reads after treatment with microDecon (deletion of reads on the basis of negative controls), while the maximum number

of reads observed in the 42 negative controls is 3 for these 16 species. Contamination can therefore be excluded during the collection and / or laboratory steps.

For each identified taxon we reported their percentages of identity with the COI reference to which they are assigned, as well as the total read number (S6 Table). This table also reports the records of these taxa in the Amazon and the Beni region.

These 252 taxa are distributed among 149 different genera in 11 orders: 45.2% of the taxa are Siluriformes, 34.5% Characiformes, 9.1% Gymnotiformes, 6.0% Cichliformes. The orders Clupeiformes, Myliobatiformes, Perciformes, Synbranchiformes, Salmoniformes, Osteoglossiformes, Cyprinodontiformes together account for the remaining 5.2% of the identified taxa.

The number of taxa per site averaged 51.03 (SD = 23.1) and ranged between 2 to 105. Overall, 53.5% of taxa were identified at four sites or less (S2 Fig). A single taxon, *Prochilodus nigricans*, was observed at all sites except Pelechuco which is located at 4,508 m a.s.l. At this site only *Orestias aff. ascotanensis* (agassizii group) and *Oncorhynchus mykiss* were identified.

A significant correlation was observed between taxa richness observed and altitude (r = 0.85, *p*-value = 0.011), with a regular pattern of increasing richness towards downstream sites (Fig 3). In rivers, taxa richness ranged from 2 at Pelechuco (4,508 m a.s.l.) to 95 in the Beni river (210 m a.s.l.). In lakes, the two most upstream have the lowest richness values (21 and 22 taxa identified in Chalalán and Santa Rosa, respectively) and the highest was observed in the Colorada lake with 105 taxa identified (see S3 Fig and S3 Table for details).

*Geographic distribution patterns of species detected in the Tuichi basin and the Beni lakes*. The hierarchical clustering of the sampled sites shows four clusters, representing four biogeographic entities that are supported by bootstrap values greater than 81% (Fig 4). The nine lakes constitute a first exclusive biogeographic entity, separating the three most-upstream lakes (Ruta, Chalalán and Santa Rosa) from the downstream lakes. The second (Alto Tuichi) brings together the four upstream sites collected on the Tuichi river, the third includes all the sites collected on the rank 3 rivers, and finally the fourth entity groups the tributaries of rank 1 and 2. The calculated coverage rates (number of observed taxa / estimated number of taxa using Jack2 estimator) for each of the four biogeographic entities (lakes, Alto Tuichi, rank 1 and 2 rivers and rank 3 rivers) range from 67.9% to 79.7% and are not significantly different from each other (*p*-value = 0.09) showing that our sampling effort is similar among these biogeographic entities and therefore allows for comparison.

In rivers, 172 taxas were identified, of which 52.3% and 32.6% belonged to Siluriformes and Characiformes, respectively. In lakes, 153 taxa were identified, including 32% of Siluriformes and 39.9% of Characiformes. As illustrated in Fig 4, Siluriformes (brown lines) had a significantly stronger occurrence (*p*-value < 0.001) in rivers than lakes whereas Characiformes (red lines) had a stronger occurrence in lakes, although this trend was not significant (*p*-value > 0.1). Taxa belonging to the Clupeiformes and Osteoglossiformes orders were detected only in lakes, whereas Cyprinodontiformes and Salmoniformes were only detected in rivers.

In the Madidi NP alone, of the 186 taxa identified, 17 taxa are common to lakes and rivers, 14 taxa identified only in lakes and 140 taxa identified only in rivers (Fig 5). Considering the four biogeographic entities established by the hierarchical analysis, their difference in terms of species composition remains strongly marked. Only 7 taxa are common to these groups. Six taxa were identified only in the Alto Tuichi group, six in the rank 3 rivers, and 54 in the rank 1 and 2 rivers group (see S7 Table for taxa lists in each area of the Venn diagram).

Between biogeographic entities the number of taxa shows a clear increase between the Alto Tuichi (46 taxa) above 700 m a.s.l and those at lower altitude (Fig 3). Below 700 m a.s.l., species diversity follows the increase in rank and therefore in size of the sub-basins, going from 98 taxa (rank 3 tributaries) to 155 taxa (rank 1 or 2 tributaries).

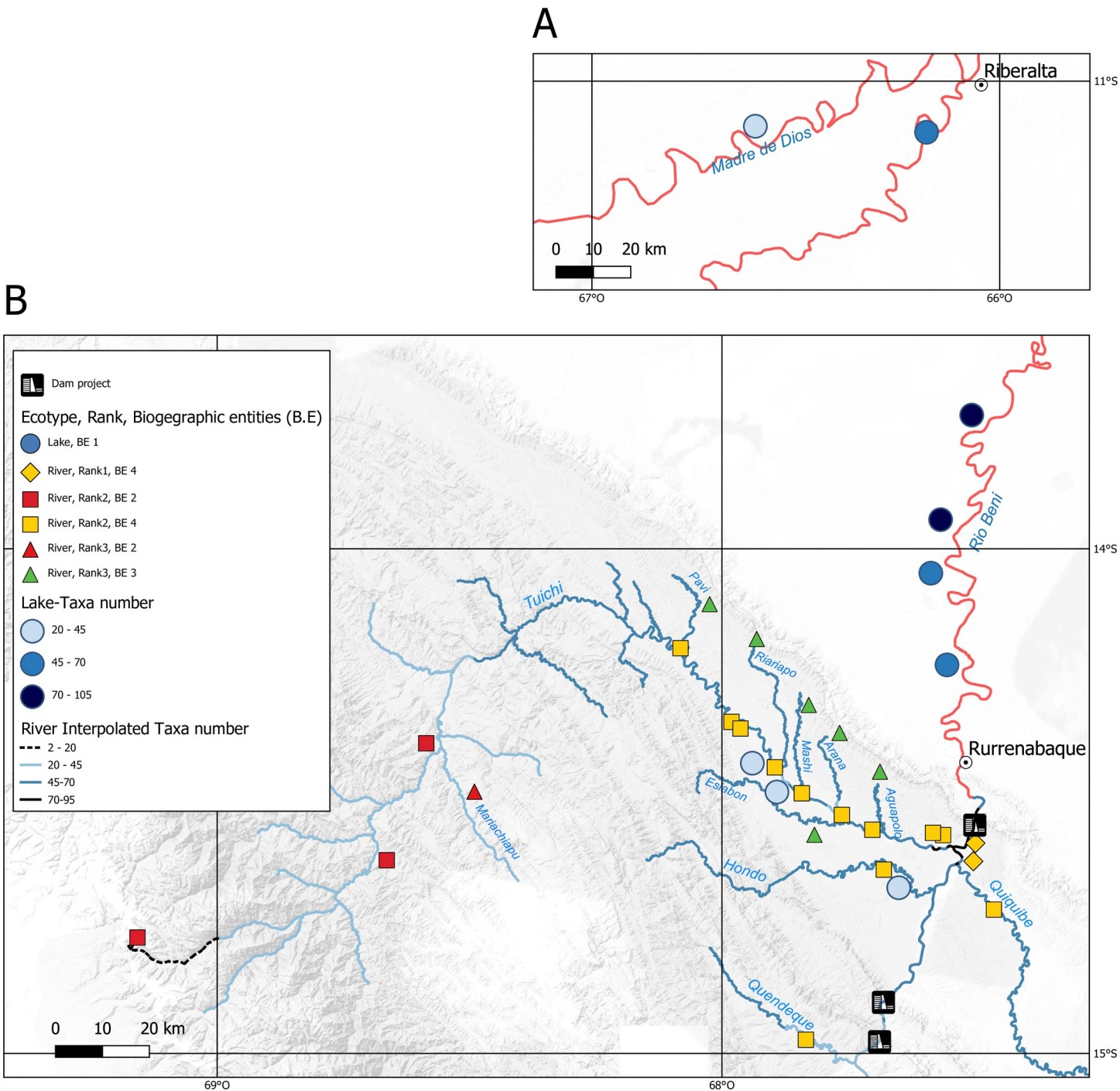

**Fig 3. Maps showing taxon richness in the study area.** For lakes (blue discs) the intensity of the colour is proportional to the observed richness (Mentiroso and Tumichucua lakes in the smaller box A). River sites are represented according to their hydrographic rank 1, 2 and 3 by diamonds, squares and triangles, respectively and a colour code associated with the biogeographic entities (BE) defined by the hierarchical analysis. The observed richness in rivers was projected and extrapolated on the hydrological network and represented by the colour code provided in the legend (network in red is not extrapolated).

*Estimation of eDNA detection distances in streams and lakes.* We assessed the minimum distance from which detectability is no longer possible in the environmental context (dry season) of our sampling. For this, we used the situation where taxa were present at a site, but not

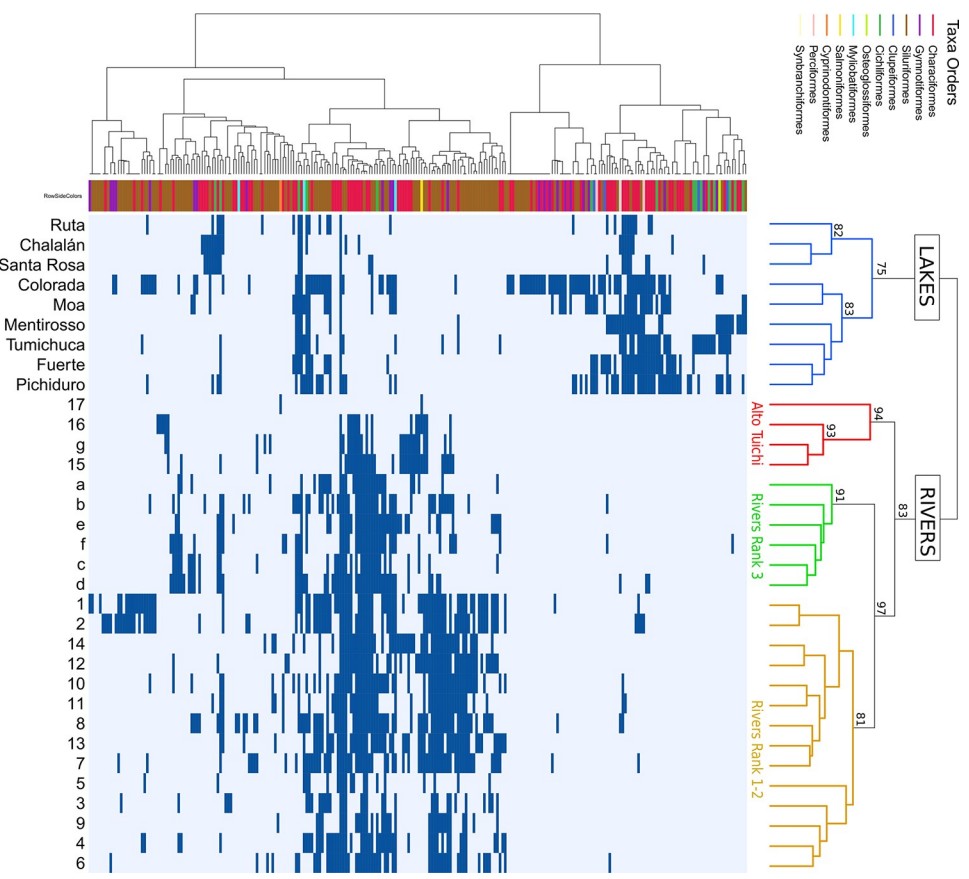

**Fig 4. Occurrence map representing the presence (blue) or absence (white) of taxa (rows) for each site (columns).**
Sites and species dendrograms are constructed using Jaccard's binary distance and a Ward clustering method. The
colours associated with the site dendrogram correspond to the biogeographic entities defined by the hierarchical
analysis. The colour associated with each leaf in the species dendrogram corresponds to its taxonomic Order. A
detailed dendrogram of the taxa with their names is available in S6 Fig. The names of the sites are those shown in Fig 1
(details in S3 Table).

detected at the site sampled immediately downstream. Of the species identified at the Arana
river site (point c on Fig 1B), 23 species are not identified at the Tuichi8 site (point 9 on Fig
1B) located only 3.7 km downstream. These 23 species have an average detection rate at the
Arana site of 71% and for 9 of these species it is 100% (i.e. identified in all Arana libraries).
Given that sampling in lakes was carried out in the dry season without water entering from the
main river, DNA detected was emitted only by species present in the lake and in its tributaries.
Therefore, in lakes, as well as in rivers, the eDNA approach provides a localized view of species
distribution by site which is estimated at a few km in our case study, as evidenced by the strong
structure of the species distribution observed among 4 clusters totally independent of the geo-
graphical distance, hence of the stream flow (Fig 3B).

*Environmental factors correlated to the biogeographic structure.* The NMDS based on Jac-
card distances between sites in fish composition had a stress value of 0.09 showing that the
representation was highly reliable. Fish populations are differentiated between ecotypes, rivers
and lakes (r2 = 0.52, *p*-value < 0.0001) (Fig 6A and S4 and S5 Figs for details) and the analysis
of variance using distance matrix confirmed a significant separation between the four biogeo-
graphic entities defined by the hierarchical clustering (*p*-value = 0.0009).

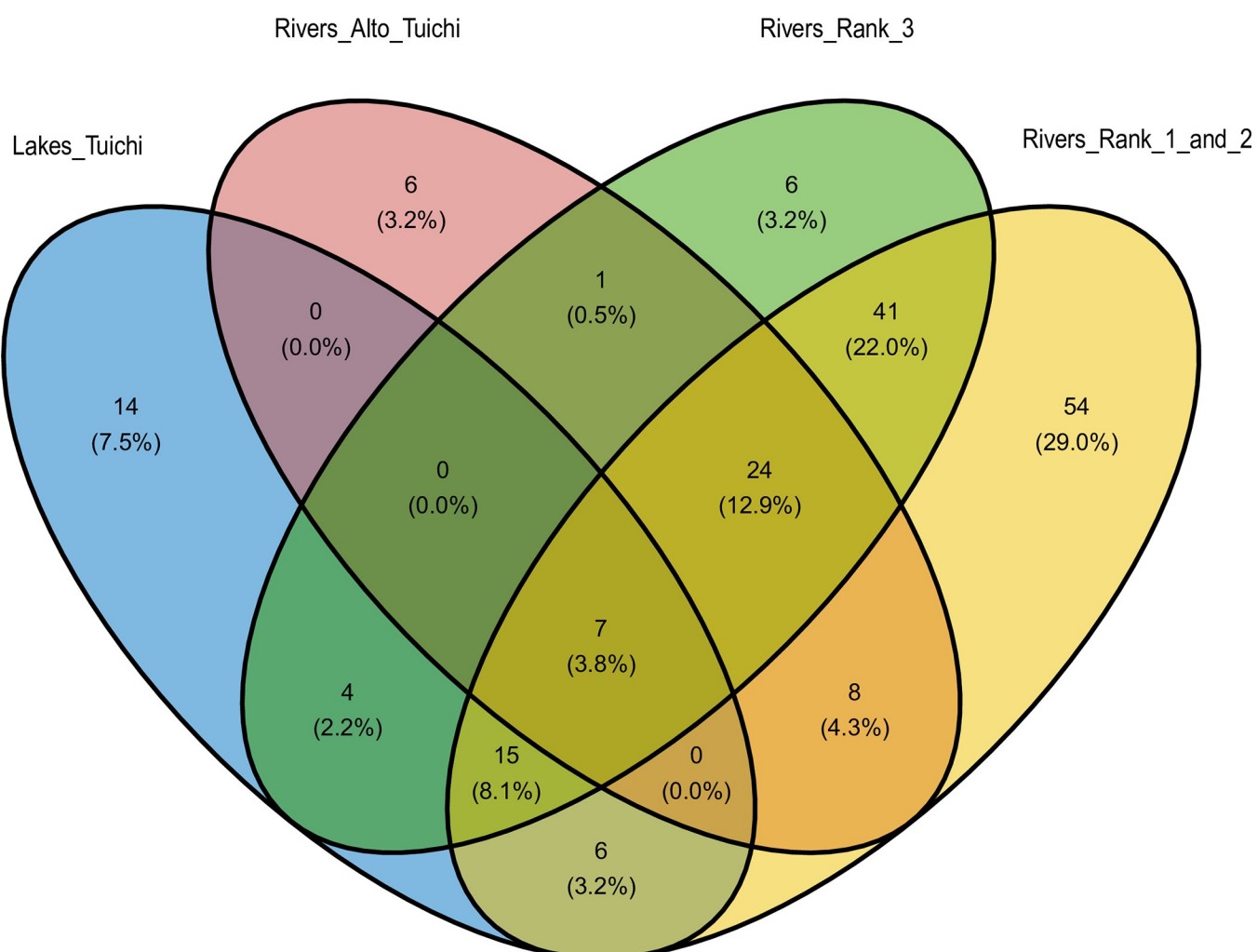

**Fig 5. Venn diagram showing the number of shared taxa between the four different biogeographic entities of the Tuichi.** Note that only lakes in Madidi NP (Chalalán, Santa Rosa, and Ruta lakes) were included in this figure. Of the 186 taxa detected in the Madidi NP alone, only 7 occur in all biogeographical entities, while 14, 6, 6 and 54 were detected only in lakes, Alto-Tuichi, rank 3 rivers and rank 1–2 rivers, respectively. The values in brackets give the percentage of the total number of taxa detected. The drawings are not to scale.

Hydro-ecoregions, longitude, altitude and pedology variables are correlated with biogeographic entities NMDS ordinations of the ichthyofauna both between rivers (Fig 6B, Table 1) and between lakes (Fig 6C, Table 1) with $p$-value $< 0.01$. Geology variation is correlated between river ichthyofauna compositions (Fig 6B, Table 1). The area of the hydrological basins and the length of the network flooding the lakes are also correlated ($p$-value $< 0.05$) with the biogeographic structure of the ichthyofauna between lakes, but the age and area of the lakes has no observed influence on their faunistic composition (Fig 6C). The Quendeque river of rank 2, which is positioned by the hierarchical classification with all other rivers of the same rank in the yellow group (Fig 4), has a position approaching the rivers of rank 3 (green group) according to the NMDS analysis (Fig 6B). This faunistic peculiarity of the Quendeque River could be explained by the small size of its hydrographic basin and network compared to other rivers of rank 2, as well as by its ecological characteristics (only site located in high humid subandean HER).

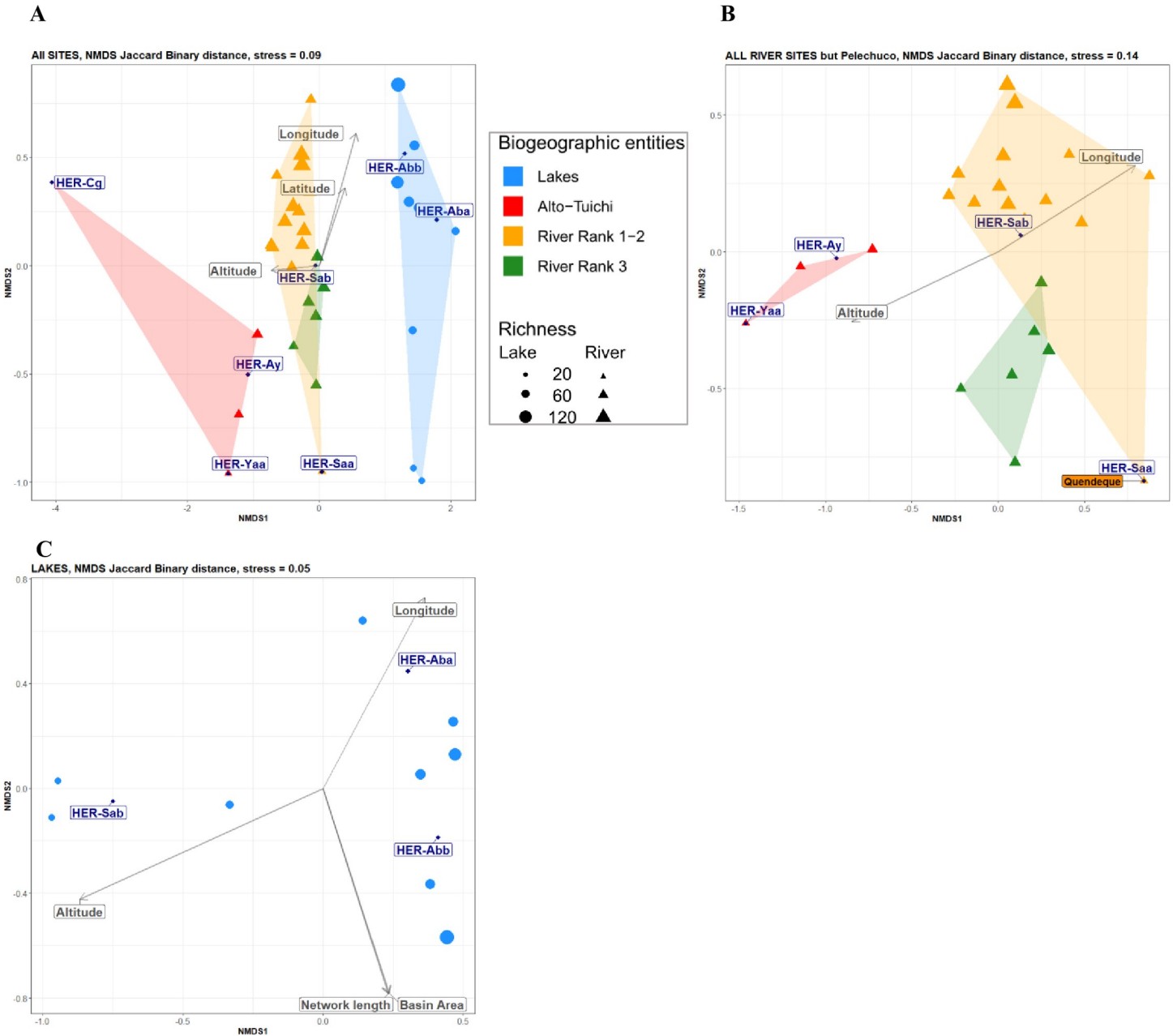

**Fig 6. NMDS ordination of the sampling sites based on the specific composition (Jaccard distance).** Ordination was performed over all sampled sites including lakes and rivers (A), then using only rivers (B) or lakes (C). The convex hulls are colour coded according to the four biogeographic entities defined by the HAC. The sizes of the dot and triangle symbols for lakes and rivers respectively, are proportional to the number of taxa observed per site. Significant (p<0.01) continuous environmental variables are represented by grey arrows. The seven hydro-ecoregions are plotted as navy blue diamond points and corresponds to Glacial mountain (Cg), High peri-Amazonian Yungas (Ya-a), Amazonian gallery forest (Ab-a), Beni Gallery Forest (Ab-b), Dry valleys of the Yungas (Ay), High humid sub-Andean (Sa-a) and Low humid sub-Andean (Sa-b).

## Discussion

### Relevance of the eDNA approach to inventory

eDNA studies carried out so far on the Neotropical ichthyofauna [25–28], although limited by the size of the reference database for the mitochondrial gene 12S used, have faced varying

**Table 1. Correlations and significance levels between environmental variables and NMDS ordinations.** These statistics are reported for the 3 NMDS ordinations carried out on i) rivers only, ii) lakes only and iii) all sites.

| | Rivers | | Lakes | | Rivers and Lakes | |
|---|---|---|---|---|---|---|
| | r2 | Pr(>r) | r2 | Pr(>r) | r2 | Pr(>r) |
| **Hydro-ecoregion HER** | 0.59 | *** | 0.81 | ** | 0.73 | *** |
| **Altitude ALT** | 0.78 | *** | 0.93 | *** | 0.53 | *** |
| **Longitude LONG** | 0.73 | *** | 0.66 | * | 0.69 | *** |
| **Latitude LAT** | | ns | 0.66 | . | 0.28 | * |
| **Geology** | 0.49 | *** | | ns | 0.61 | *** |
| **Pedology** | 0.31 | * | 0.76 | ** | 0.59 | *** |
| **Ecotype** | | na | | na | 0.52 | *** |
| **Length of the hydrographic network HNL** | | na | 0.67 | * | | na |
| **Area of the basins** | | ns | 0.67 | * | | na |
| **Age of the lakes** | | na | | ns | | na |
| **Area of the lakes** | | na | | ns | | na |

Significance codes: 0

'***' 0.001

'**' 0.01

'*' 0.05 '.' 0.1 ' ' 1

degrees of success. Jackman et al. (2021) identified 84 molecular taxonomic units (MOTU), of which only 4 to the species level (1.6%), out of the 258 fish species detected using traditional sampling methods in the Adolpho Ducke Forest Reserve near Manaus in the Brazilian Amazon. Sales et al. (2021) recovered 34 out of the 111 species (30.1%) described in the Jequitinhonha River basin, Eastern Brazil; de Santana et al. (2021) recovered 58 out of 443 species (13.1%) in the Javari River, Western Brazil; whereas up to 65% (132 out of 203 species) were detected in several watercourses of French Guiana [26]. Since the beginning of the 19th century until recently, 444 species of fish have been identified in the Beni basin [1]. Our eDNA approach, although carried out in a small portion of the same basin (10% of the area of the Beni basin) and during a single hydrological period, has resulted in a list of 207 species (47%). Among these species, 57 are newly identified for the Beni watershed as compared to the current most exhaustive census compiled by [1] among which 16 were named "affinis" because they were molecularly close to species not described in the Amazon basin. These 16 species could potentially be new species for science. All these results confirm our hypothesis that eDNA would be more efficient than traditional assessment methods to inventory species diversity and to unravel rare or cryptic species. They also emphasize the need to fill the knowledge gaps in the Tuichi sub-basin [2], where species potentially new for science were detected, which should foster further sampling efforts using both traditional and eDNA approaches.

In accordance with recent recommendations [58], which identified 87% of expected fish fauna in small streams with a single 34 l sample, we filtered an average of 38 l of water per site which allowed us to collect sufficient eDNA and achieve a mean coverage of 79.8% of the estimated number of species under the limits of the available databases (Bold, GenBank). Although increasing the sequencing effort could significantly increase the observed species diversity as shown by [59], sequences rarefaction tests performed on our samples show that this effort was sufficient (S1A Fig).

It is worth noting that the results presented here were achieved in a 4-week sampling timeframe, for an average analysis and sequencing cost of approximately $450 per site (year base 2020), further emphasizing the effectiveness of the eDNA approach to inventory species. This

is an extremely low cost compared to that of sampling specimens over decades, and which has enabled us to increase the species inventory by 11.4% in a watershed that has been inventoried for over a century [1]. Some of these newly detected species could be cryptic or very scarce, or new species for science but not yet described because they are morphologically similar to already described species: for example, the species of the genus *Prochilodus* noted affinis in this paper, and probably already identified by metabarcoding of bulks of larvae further downstream from our study area [11].

**A limited detectability of eDNA in the Amazon region.** Our study showed that some species that were detected at a sampling site were no longer detected a few kilometres downstream (around 3 km) at the next sampling site. However, it should be noted that the sites where these species were identified with their DNA do not necessarily correspond to their locations, so this distance may be underestimated. Although it is difficult to compare diffusion distances between different studies, which depends on detectability and therefore on the experimental set-up (volume of water filtered, fraction of DNA analysed, etc.), Civade et al. (2016) [60] estimated diffusion distances in French Guiana at between 2 and 3 km, similar to those observed here. However, in the Rhône in France, eDNA was found between a few km to 130 km from its emitting source, but with a detection rate below 0.4% for the latter [61].

A hypothetical explanation could be that the detectability of fish species in the Amazon basin could be lower than in other regions. As the specific diversity in the Amazon is much greater than that observed at temperate latitudes, if the biomasses are similar, the eDNA concentration for each Amazonian species would therefore be lower, reducing their detectability.

In our case, *Orestias ascotanensis*, present at the extreme upstream Pelechuco site (a creek of less than 1 m width), was not detected at the Virjen site located 61.7 km downstream (a river over 100 m wide) even though eDNA could drift down this distance in less than 24 hours given that the speed of current at Virjen was around 0.7 m/s.

It is therefore difficult to attribute this lack of detection to a total degradation of DNA, given that the half-life of DNA molecules in the aquatic environment is generally considered to be several days, even in tropical conditions [62, 63].

An additional explanation could be the increase in flow rate which, by increasing eDNA dilution, would decrease detectability as highlighted by [64] who showed the influence of both temperature and stream flow level on *Corbicula fluminea* (clam) detectability. In our case study, both the high water temperature (mean over sites = 25˚C, SE = 4.9) and the large and rapid increase in river discharge, are likely to limit the detectability of eDNA a few kilometres downstream of its emission. We can also assume that the length of our barcodes (185–285 bp), which is greater than those sometimes used, such as 12S (65 bp), would allow us to target less degraded DNA, which has been recently released into the environment and has not diffused very much [65]. The fact that 73 taxa were only detected by the MK marker (185 bp) compared to the 17 detected only by MK2 (285 bp), tend to lend support to this hypothesis. The use of long barcodes might thus provide a more local image of the diversity detected.

The only exception to this trend is the rainbow trout, *Onchorhynchus mykiss*, a freshwater Salmoniformes, that was unexpectedly detected in several sites of the Tuichi and Beni basins (S7 Table), usually with high identity (>98%). Given what was just explained about the limited diffusion of eDNA, this species, which lives in cold waters at high altitude should not be encountered in the lower portion of the Tuichi River, nor in the Beni River. A likely explanation, however, is the presence of many trout aquaculture facilities in the Andean portion of the Tuichi and in several of its tributaries. The presence of trout in very large, unnatural, concentration, as is the case in intensive aquaculture, could release large amounts of DNA in the waters that could disperse downstream in spite of the above-mentioned factors limiting eDNA diffusion.

*Biogeographic structure.* The fish species composition revealed among the sites of this study through environmental DNA provides a sufficiently localised signature for a biogeographic pattern to emerge. The sampling sites were close to the confluences of the Quiquibey, Quendeque and Hondo rank 2 rivers and the rank 3 tributaries of the Tuichi, allowing the identification of the fauna located at only few km upstream of the sampling points. In oxbow lakes and tectonic lakes, the absence of inflows from streams allows the identification of the local ichthyofauna present at the sampled site. Four biogeographic entities are highlighted according to the specific composition of their ichthyofauna in relation to the structure of the hydrographic network (rank 1–2, rank 3, Alto-Tuichi) with an even stronger distinction of the ichthyofauna between rivers or lakes biotopes constituting the fourth entity. The ability of eDNA to more accurately report on the structuring of species than more traditional inventories (net, electric fishing, rotenone) is recognized [26, 66]. The entire ecosystem is better apprehended with a sharp reduction in the biases inherent to sampling a complexity of habitats in which rare and cryptic species would be much more difficult to access and register by traditional means. Similar richness coverage between the 4 biogeographic entities (from 70 to 80%) reflecting similar sampling efforts allows comparing their ichthyodiversity with limited bias.

The distribution patterns of the ichthyofauna according to the ranks of the tributaries and therefore according to their sizes, have been demonstrated many times in all latitudes (review in Hugueny, Oberdorff, & Tedescco, 2010 [67]). In the Beni basin, the patterns of differentiation between lake and river ichthyofauna are particularly contrasting with all the lakes from upstream to downstream of the Beni clustered in the same biogeographic entity. Furthermore, two sub-units can be distinguished among lakes: the lakes of the Rio Beni plain and those located in the upstream portion of the basin in the Madidi reserve (lakes Ruta, Santa Rosa and Chalalán). For Santa Rosa and Chalalán lakes, which are tectonic and older, species diversity is lower than in the oxbow lakes, however, for the latter neither their age of appearance, nor their surface area is correlated with their species richness. We also note that lake Ruta, although young (2005) and small (4 ha), has the same number of species as the older (<1950) and much larger (180 times) lake Mentiroso located in the Madre de Dios plain (Fig 3). In contrast with the Ruta lake, Mentirosso is under strong human pressure (fishing, mining activities).

The distribution of fish populations' biogeographic entities in the Tuichi basin is strongly correlated with hydro-ecoregions patterns [31] that include several local environmental variables, sometimes correlated with each other, highlighting the relationship between fish distribution and waters quality in the Amazon [68]. The physico-chemical parameters involved in water quality have not been directly measured in this study, but are nevertheless integrated in the four main categories of variables considered. The biogeography of the lake ichthyofauna is correlated to altitude, itself correlated to the longitude and the nature of soils. The composition of the ichthyofauna of the different rivers is correlated with the same environmental parameters as lakes, to which is added geology, itself in relation to altitude. Usually, the nature of the substrate on which the continental waters flow (river and lake) partly explains the distribution of the Amazonian ichthyofauna [69], since the variation of the soils and subsoils drives the variation of the physicochemical parameters of the water, the living environment of the fish, as well as the type of riparian vegetation, which is itself a food resource and nursery for numerous fish species [70].

In the present study the observed species richness drops considerably with altitude in lotic biotopes, with less than a dozen species detected only above 700 m elevation. The decline in species richness with increasing altitude is described as being related to decreased temperature, an increase in dissolved oxygen and pH, also accompanied by a turnover of species, as observed in the Colombian Andes [71]. However, this drop in diversity is also accompanied by an increase in the proportion of endemic species in the Andean foothills: around 40% of the

400 to 600 species inventoried, including species that are threatened or already extinct [72]. Moreover, this increase in endemicity with altitude would correspond to a diversification of the functional groups distributed in the basin headwaters, generating an increase in beta diversity [73].

*Human threats and conservation*. Many watersheds originating in the Andean valleys are under increasing anthropogenic pressure due to the construction of dams, gold mining [74] and pollution by wastewater from cities and industries (e.g. construction of sugar mills close to sites such as the Fuerte lake). In order to set up conservation strategies, the biodiversity of the aquatic fauna distributed in particular in the Andean foothills should be generalised and investigated rapidly, as we have done for the Tuichi basin, using eDNA approaches. Owing to its location directly downstream of the capital city, La Paz, and its anthropogenic pressure (mining, pollution, landscape degradation, exploitation; [11]), but also to the presence of the Madidi NP and knowledge gaps [1], the Beni basin should be considered a priority area for carrying out further eDNA surveys.

The results and methodology of the present study could serve as a reference point for future comparisons of river basins under different anthropogenic pressures with the lightly anthropized Andean basin of the Tuichi River. On a larger scale, anthropogenic effects in the Beni watershed, and in particular fishing pressures that remain poorly documented [75], should be estimated and related to the specific assemblages and richness observed. An important anthropogenic threat is the introduction of species, such as *Arapaima gigas*, species CITES II on all its natural distribution range. Naturally absent from the upper Madeira, it has been introduced in the 70s from an aquaculture facility in Lake Sandoval on the Madre de Dios River, tributary of the Beni River [76, 77]. We have detected *Arapaima* in four lakes, Mentirosso, Tumichuca, PichiDuro and Colorado, but not higher upstream. The lake Colorado would therefore be at the front of colonisation of the species on the Beni River at the sampling period. The species *Prochilodus argenteus*, is normally endemic to the San Francisco basin [78]. A molecularly close species (COI gene), named *Prochilodus* aff. *argenteus* had been identified in the Beni River using metabarcoding of fish larvae [11]. More recently [1], reported its occurrence in the Coari River in the Brazilian Amazon. Now, using eDNA, we have again detected this species in the Beni watershed, which suggests a wider distribution range than currently described. However, as this economically important species has already been introduced elsewhere in Brazilian rivers [25], an introduction in the Beni watershed is also likely. Another growing anthropogenic threat is the rapid expansion of hydroelectric dams in the Amazon basin [79–83]. Three dams are currently planned in our study area (Fig 1), between the protected areas of the Madidi NP: one at the Bala strait in the middle of our study area, and two slightly upstream of it at the Chepete and Beu straits [11]. Their construction would lead to > 50% losses of network connectivity in the Beni basin [79]. It would also result in profound alterations in the sedimentation patterns of the river (volume and flow of sediments), homogenizing downstream habitats in the river and in the lakes [84]. Additionally, these dams would also result in the formation of extensive upstream reservoirs and the Bala dam would likely permanently flood a large portion of our study area. Other foreseeable impacts of these dams are similar to that observed on riverine fish communities elsewhere in the world and range from local extinction of species that cannot adapt to reservoirs and dam-tailwater conditions, to basinwide extirpations of migratory species [85, 86]. The Amazon basin hosts a rich migratory fish fauna that accounts for the bulk of fisheries landing and dam construction is expected to severely impacts migratory species (review in Duponchelle et al., 2021 [87]). Several of these migratory species holding commercial and cultural importance were identified in our study area. A spawning area for several of these species, threatened by anthropogenic activities such as arid extraction or pollution, was recently identified in our study area [11]. The alteration of

local environmental conditions by damming (sediment and flow patterns modifications, permanent flooding of large area) will likely further compromise the usefulness of the identified spawning area. It will profoundly modify the structure of fish communities and the fisheries that depend on them, as observed in other dammed rivers of the Amazon basin (review in Duponchelle et al., 2021 [87]).

*Contributions and limitations of the eDNA approach in the biogeographic study of the region.* The eDNA approach permitted a study of fish biodiversity in the Tuichi sub-basin and broader Beni basin within and immediately adjacent to the Madidi NP with a high species diversity thanks to a high taxonomic resolution, since 93.3% of the assigned OTUs (5,246) are at the species level. A majority of OTUs (8,205) although aligned with at least one sequence in the COI fish database could not be assigned taxonomically due to a too low percentage of identity with the reference (<97%). This is closely related to the fact that of 2,406 known Amazonian fish species [2], only a third (694) have a COI barcode available. The effectiveness of the method would therefore be considerably enhanced if the reference databases used for taxonomic attribution (Bold, GenBank) were improved over time, thereby improving our species identification capabilities.

Markers other than COI have been used to record fish biodiversity, such as a 12S marker in French Guiana [26]. But the smaller size of the latter (65 bp versus 185–285 bp for the COI barcodes used here) and an even more limited database (181 available 12S barcode in Cilleros et al., 2018 [26]) than that of the COI, makes this marker less efficient than the COI, with more difficulty in assigning OTUs below family or gender level. The combination of several markers [88] and the availability of more exhaustive reference databases would allow a more accurate appreciation of species diversity. The eDNA approach, already widely used for species inventories, also seems to be a promising tool for localised studies to assess the impacts of anthropogenic disturbance on biodiversity, such as dams, factories or mines, but also for carrying out a rapid census of biodiversity on more extensive regional scales. Ecological and conservation studies will require the complementarity of traditional sampling approaches to provide information on recruitment, age, size and precise abundance, as well as to identify the specimens needed to complete the DNA reference databases. Nevertheless, eDNA approaches to access large-scale species inventories in a rapid and non-destructive manner can complement these studies. Quantitative approaches based on eDNA sampling are being developed using qPCR or digital PCR techniques, in order to gain access to relative biomasses [89, 90]. Although many technical or biological biases still need to be overcome, access to biomass would provide information on the demographic patterns of species required to put in place conservation strategies essential for sustainable ecosystem management. Fishes are among the freshwater biota most threatened by global changes [91, 92]. The evolution of fish biodiversity in the Tuichi study area could be monitored over several years in relation to climate change and dynamic anthropogenic threats in the region, and serve as a reference for the study of other river basins in the Amazon.

## Supporting information

**S1 Fig. Rarefaction curves at library and site level.**
(PDF)

**S2 Fig. Frequency spectrum of taxa.**
(PDF)

**S3 Fig. Taxonomic diversity at order level per site.**
(PDF)

**S4 Fig. NMDS ordination (site names plotted).**
(PDF)

**S5 Fig. NMDS ordination including all sites.** Orders are split into different facets.
(PDF)

**S6 Fig. Species dendrogram (binary Jaccard distance and Ward2 aggregation).**
(PDF)

**S1 Text. Scripts and command lines used for data analysis and plotting figures.**
(PDF)

**S2 Text. Assessment of the taxonomic resolution of markers used on Amazonian fish.**
(PDF)

**S1 Table. Informations on sites and points of collect (date, collected volume, depth, coordinates).**
(XLSX)

**S2 Table. COI fish database used (fasta file).**
(XLSX)

**S3 Table. Environmental variables (soils, altitude, coordinate, geology, richness, Jack2. . .).**
(XLSX)

**S4 Table. Sequencing results per library.**
(XLSX)

**S5 Table. Taxa abundance per library.**
(XLSX)

**S6 Table. Quality of assignation (%ID) and list of species assigned to affinis status.**
(XLSX)

**S7 Table. List of taxa present in each area of the Venn diagram constructed at the cluster level.**
(XLSX)

## Acknowledgments

The authors acknowledge the IRD itrop HPC (South Green Platform) at IRD Montpellier for providing HPC resources that contributed to the research results reported within this paper (URL: https://bioinfo.ird.fr/- http://www.southgreen.fr).

## Author Contributions

**Conceptualization:** Cédric Mariac, Fabrice Duponchelle, Jean-François Renno.

**Formal analysis:** Cédric Mariac, Fabrice Duponchelle, Jean-François Renno.

**Methodology:** Cédric Mariac, Guido Miranda, Camila Ramallo, Gabriel Tarifa, Jean-François Renno.

**Validation:** Cédric Mariac.

**Writing – original draft:** Cédric Mariac, Fabrice Duponchelle, Jean-François Renno.

**Writing – review & editing:** Cédric Mariac, Fabrice Duponchelle, Guido Miranda, Camila Ramallo, Robert Wallace, Gabriel Tarifa, Carmen Garcia-Davila, Hernán Ortega, Julio Pinto, Jean-François Renno.

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
