## [Decision Letter · Decision Letter 0]

7 Oct 2021

PONE-D-21-19821Unveiling biogeographical patterns of the ichthyofauna in the Tuichi basin, a biodiversity hotspot in the Bolivian Amazon, using environmental DNA.PLOS ONE

Dear Dr. Mariac,

Thank you for submitting your manuscript to PLOS ONE. After careful consideration, we feel that it has merit but does not fully meet PLOS ONE’s publication criteria as it currently stands. Therefore, we invite you to submit a revised version of the manuscript that addresses the points raised during the review process.

I got the recommendations and comments from an expert reviewer on the field. The reviewer agreed that the manuscript is technically sound and the data support the conclusions.However, lack of the explanation in Methods and Results sections were suggested by the reviewers, and I totally share their comments. Therefore, I can invite you to submit a revised version of the manuscript that addresses the points raised by the reviewer.

We look forward to receiving your revised manuscript.

Kind regards,

Hideyuki Doi

Academic Editor

PLOS ONE

Journal Requirements:

2. In your Methods section, please provide additional location information, including geographic coordinates for the data set if available.

“This study was financed by IRD and the Wildlife Conservation Society. The LMI EDIA and UMR DIADE financed the metabarcoding analyses. The authors acknowledge the IRD itrop HPC (South Green Platform) at IRD Montpellier for providing HPC resources that contributed to the research results reported within this paper (URL: https://bioinfo.ird.fr/- http://www.southgreen.fr).”

“YES-This study was financed by IRD and the Wildlife Conservation Society. The LMI EDIA and UMR DIADE financed the metabarcoding analyses.”

6. We note that Figure 1 in your submission contain [map/satellite] images which may be copyrighted. All PLOS content is published under the Creative Commons Attribution License (CC BY 4.0), which means that the manuscript, images, and Supporting Information files will be freely available online, and any third party is permitted to access, download, copy, distribute, and use these materials in any way, even commercially, with proper attribution. For these reasons, we cannot publish previously copyrighted maps or satellite images created using proprietary data, such as Google software (Google Maps, Street View, and Earth). For more information, see our copyright guidelines: http://journals.plos.org/plosone/s/licenses-and-copyright.

a. You may seek permission from the original copyright holder of Figure(s) [#] to publish the content specifically under the CC BY 4.0 license.

We recommend that you contact the original copyright holder with the Content Permission Form (http://journals.plos.org/plosone/s/file?id=7c09/content-permission-form.pdf) and the following text: “I request permission for the open-access journal PLOS ONE to publish XXX under the Creative Commons Attribution License (CCAL) CC BY 4.0 (http://creativecommons.org/licenses/by/4.0/). Please be aware that this license allows unrestricted use and distribution, even commercially, by third parties. Please reply and provide explicit written permission to publish XXX under a CC BY license and complete the attached form.”

b. If you are unable to obtain permission from the original copyright holder to publish these figures under the CC BY 4.0 license or if the copyright holder’s requirements are incompatible with the CC BY 4.0 license, please either i) remove the figure or ii) supply a replacement figure that complies with the CC BY 4.0 license. Please check copyright information on all replacement figures and update the figure caption with source information. If applicable, please specify in the figure caption text when a figure is similar but not identical to the original image and is therefore for illustrative purposes only. The following resources for replacing copyrighted map figures may be helpful:

USGS National Map Viewer (public domain): http://viewer.nationalmap.gov/viewer/ The Gateway to Astronaut Photography of Earth (public domain): http://eol.jsc.nasa.gov/sseop/clickmap/ Maps at the CIA (public domain): https://www.cia.gov/library/publications/the-world-factbook/index.html and https://www.cia.gov/library/publications/cia-maps-publications/index.html NASA Earth Observatory (public domain): http://earthobservatory.nasa.gov/ Landsat: http://landsat.visibleearth.nasa.gov/ USGS EROS (Earth Resources Observatory and Science (EROS) Center) (public domain): http://eros.usgs.gov/# Natural Earth (public domain): http://www.naturalearthdata.com/.

Additional Editor Comments (if provided):

I got the recommendations and comments from an expert reviewer on the field. The reviewer agreed that the manuscript is technically sound and the data support the conclusions.However, lack of the explanation in Methods and Results sections were suggested by the reviewers, and I totally share their comments. Therefore, I can invite you to submit a revised version of the manuscript that addresses the points raised by the reviewer.

Reviewers' comments:

Reviewer's Responses to Questions

**Comments to the Author**

1. Is the manuscript technically sound, and do the data support the conclusions?

Reviewer #1: Partly

2. Has the statistical analysis been performed appropriately and rigorously? 

Reviewer #1: Yes

3. Have the authors made all data underlying the findings in their manuscript fully available?

Reviewer #1: Yes

4. Is the manuscript presented in an intelligible fashion and written in standard English?

Reviewer #1: Yes

5. Review Comments to the Author

Reviewer #1: Review

Comments to the Author

Environmental DNA (eDNA) approaches have been revolutionizing the assessment and monitoring of biodiversity in both aquatic and terrestrial ecosystems, yet their applications to rich biodiversity sites, especially in the Neotropical region, have largely been limited despite their potential. In this study, the authors assessed the potential of eDNA approaches for monitoring Neotropical fishes by analyzing eDNA from water samples collected both in rivers and lakes in the Bolivian Amazon region. Authors amplified DNA using a newly designed two COI metabarcoding primers targeting Amazon fishes and sequenced the amplicons on Novaseq illumina platform. Thus, they detected a high fish species richness and described their biogeographical distribution patterns related it to a set of ecological descriptors.

The topic of this manuscript fits to the scope of PlosOne, and it is timely and may potentially attract readers in both the field of Ichthyology and of biodiversity assessment using eDNA techniques. However, I found several points (detailed below), especially in the Method section that the authors should address to improve the clarity of the manuscript.

Abstract

Abstract is over the 300 words limit.

Although authors should explain how the study was done without methodological detail, it would greatly strengthen the manuscript to inform in the Abstract the use of designed primers targeting Amazon fishes.

Introduction

L60-61: “The Tuichi basin is included in the Madidi National Park and Natural Area of Integrated Management (hereafter abbreviated Madidi NP), a recognised biodiversity hotspot [8–10] and also likely hosts one of the few identified spawning area for many fish species [11]”

It is not recognized among the world’s 36 hotspots biodiversity. If it is only locally/nationally recognized I would suggest replacing the term “biodiversity hotspot” to a “rich biodiversity site”.

L68-86: Here authors include a brief review of the use of eDNA techniques including examples on amphibians, aquatic mammals, and fishes. However, they failed to current state of eDNA research in the Amazon region. I strongly suggest that authors strengthen this part. A key recent literature on the Amazon fish (Jackman et al. 2021) and mammals (Sales et al. 2019) might help.

L92-94: I would state specifically what your objectives were. - Why did you do this using eDNA? What were your objectives? The aim of the study as it is, seems to be a report case. Make clear the main aims of the study and how significant they are for the field.

Methods

What is the average distance between the 34 collection sites?

Did authors collect eDNA samples across years (2017, 2018, 2019) in the same 34 sites?

L129-130: Include the License/authorization Number. Permits and approvals obtained for the work, including the full name of the authority that approved the study; if none were obtained, authors should explain why.

eDNA field sampling

Were temperature and pH recorded at each site?

Inform depth of sample water collection.

Detailed information on the field and laboratory negative controls are missing. Make clear what is empty control. Please describe what was the field, “alien” and laboratory negative controls (commercial water, distilled water, or ultrapure water?). Additionally, please describe at which site and at which timing the field negative controls were collected and filtered. Also, detailed information on the sequencing platform is also required. I would guess that you used Illumina NovaSeq 6000, but please make it clear.

Did authors validate the designed primers and laboratory protocols before eDNA extraction?

How did authors verify taxonomic resolution of COI fragments targeted by the designed primers? Did they use a database of Neotropical fishes to improve taxonomic assignments? With how many species?

Results

303-307: It is not clear how authors discarded the four taxa.

The accession number AMNHI782-12 (Marcusenius monteiri) was not found in GenBank. Make sure it is correct. And the other three taxa have their information matched in both GenBank and Boldsystem (but Epinephelus marginatus JX124778.1 was found in Bolsystem by its voucher LBPV53018).

How many reads do those four discarded taxa have? How is the author sure they were not putative contamination?

L315-316: Did authors remove the maximum number of reads found in the controls from all samples before downstream the analysis to avoid putative contamination?

L323-324: It is a very low threshold to ensure non-contamination. Most eDNA studies remove all MTOUS <10 reads from the final dataset. Do authors have a citation for why 4 reads were adopted or just type a further clear justification.

L332-333: One of the cost-efficiency of using eDNA is to detect rare or cryptic species that are difficult to detect by using traditional methods. However, it is highly prone to putative contamination. Since key information on the Methods section is missing, I am not truly convinced that these 16 named affinis are not putative contamination. How many reads do those taxa have? Is it possible they have been introduced in the Amazon basin?

Discussion

Authors should put their results in the context with respect to other eDNA studies on fish surveys in the Amazon and Neotropical region. It would greatly strengthen the discussion to briefly detail the survey success obtained in this study to the success of other studies.

Authors should discuss in more detail the conservation issues regarding the species found in the study site, especially regarding the newly detected species.

Are there any threatened species? Is it possible for any of those affins species to be an exotic species introduced in the area? Prochilodus argenteus, for example, is an invasive species in several Brazilian river basins. Is it possible for the species to be present in the study site?

English

The level of English is very high throughout the proposal, but there are some minor errors throughout. It would therefore be worthwhile having the final version carefully reviewed before final submission.

Figure 3: see comments in the pdf file.

Figure S3: Taxonomic diversity at Order level per site: Axis is very difficult to read. I suggest increasing the text size, or include this figure in an entire page.

Minor comments are in the main text pdf file. Good luck revising the manuscript and I look forward to seeing the revised version!

6. PLOS authors have the option to publish the peer review history of their article (what does this mean?). If published, this will include your full peer review and any attached files.

Reviewer #1: No

---

## [Author Response · Author response to Decision Letter 0]

10 Dec 2021

PONE-D-21-19821

Unveiling biogeographical patterns of the ichthyofauna in the Tuichi basin, a biodiversity hotspot in the Bolivian Amazon, using environmental DNA.

Answer : Done

2. In your Methods section, please provide additional location information, including geographic coordinates for the data set if available.

Answer: Details on geographic locations of sampling sites are reported in Table S1.

Answer : rephrased as follow :

“The Ministerio de Medio Ambiente y Agua (MMAyA) of Bolivia approved the study and provided permits (authorization numbers: FCPV-IE-0196/2017; FCPV-IE-0030/2018; FCPV-IE-0064/2019) for collection[A1] , export and analysis of the filter cartridges to the French National Institute of Research for Sustainable Development (IRD) laboratories in France.”

Done see point 5.

“This study was financed by IRD and the Wildlife Conservation Society. The LMI EDIA and UMR DIADE financed the metabarcoding analyses. The authors acknowledge the IRD itrop HPC (South Green Platform) at IRD Montpellier for providing HPC resources that contributed to the research results reported within this paper (URL: https://bioinfo.ird.fr/-http://www.southgreen.fr).”

“YES-This study was financed by IRD and the Wildlife Conservation Society. The LMI EDIA and UMR DIADE financed the metabarcoding analyses.”

Answer : statements of changes are mentioned in our cover letter and also reported below so that you can change the online submission form on our behalf.

Funding:

CGD has been supported by Programa Nacional de Investigación Científica y Estudios Avanzados (PROCIENCA) [grant number 017-2018-FONDECYT]. GM, RW, GT, CR have been supported by Gordon and Betty Moore Foundation [grant number GBMF331.07]. CM, FD, JFR, CDG have been supported by Laboratoire Mixte International - Evolution et Domestication de l’Ichtyofaune Amazonienne (No grant number). The funders had no role in study design, data collection and analysis, decision to publish, or preparation of the manuscript.

 Acknowledgments:

The authors acknowledge the IRD itrop HPC (South Green Platform) at IRD Montpellier for providing HPC resources that contributed to the research results reported within this paper (URL: https://bioinfo.ird.fr/-
http://www.southgreen.fr).

6. We note that Figure 1 in your submission contain [map/satellite] images which may be copyrighted. All PLOS content is published under the Creative Commons Attribution License (CC BY 4.0), which means that the manuscript, images, and Supporting Information files will be freely available online, and any third party is permitted to access, download, copy, distribute, and use these materials in any way, even commercially, with proper attribution. For these reasons, we cannot publish previously copyrighted maps or satellite images created using proprietary data, such as Google software (Google Maps, Street View, and Earth). For more information, see our copyright guidelines:http://journals.plos.org/plosone/s/licenses-and-copyright.

a. You may seek permission from the original copyright holder of Figure(s) [#] to publish the content specifically under the CC BY 4.0 license.

We recommend that you contact the original copyright holder with the Content Permission Form (http://journals.plos.org/plosone/s/file?id=7c09/content-permission-form.pdf) and the following text: “I request permission for the open-access journal PLOS ONE to publish XXX under the Creative Commons Attribution License (CCAL) CC BY 4.0 (http://creativecommons.org/licenses/by/4.0/). Please be aware that this license allows unrestricted use and distribution, even commercially, by third parties. Please reply and provide explicit written permission to publish XXX under a CC BY license and complete the attached form.”

Answer : A Permission to publish Content under CC-BY License was obtained from the publisher IAHS and the completed Content Permission Form uploaded.

The figure 1 caption using the copyrighted figure have been implemented with the following sentence :

“Reprinted from Wasson et al. 2002 under a CC BY license, with permission from International Association of Hydrological Sciences, original copyright 2002.”

---

## [Editor Report · Decision Letter 1]

22 Dec 2021

Unveiling biogeographical patterns of the ichthyofauna in the Tuichi basin, a biodiversity hotspot in the Bolivian Amazon, using environmental DNA.

PONE-D-21-19821R1

Dear Dr. Mariac,

We’re pleased to inform you that your manuscript has been judged scientifically suitable for publication and will be formally accepted for publication once it meets all outstanding technical requirements.

Kind regards,

Hideyuki Doi

Academic Editor

PLOS ONE

Additional Editor Comments (optional):

I carefully checked the revised manuscript as well as the response letter. I agree the revisions according to the reviewers’ comments and now can recommend to publish the paper in this journal.
---

## [Editor Report · Acceptance letter]

26 Dec 2021

PONE-D-21-19821R1 

Unveiling biogeographical patterns of the ichthyofauna in the Tuichi basin, a biodiversity hotspot in the Bolivian Amazon, using environmental DNA. 

Dear Dr. Mariac:

I'm pleased to inform you that your manuscript has been deemed suitable for publication in PLOS ONE. Congratulations! Your manuscript is now with our production department. 

Kind regards, 

on behalf of

Dr. Hideyuki Doi 

Academic Editor

PLOS ONE